# Variations in the Course and Diameter of the Suprascapular Nerve: Anatomical Study

**DOI:** 10.3390/ijerph19127065

**Published:** 2022-06-09

**Authors:** Marta Montané-Blanchart, Maribel Miguel-Pérez, Lourdes Rodero-de-Lamo, Ingrid Möller, Albert Pérez-Bellmunt, Carlo Martinoli

**Affiliations:** 1Unit of Human Anatomy and Embryology, Unity of Histology, Department of Pathology and Experimental Therapeutics, Faculty of Medicine and Health Sciences (Bellvitge Campus), University of Barcelona, 08907 Barcelona, Spain; marta.montane@eug.es; 2Physiotherapy Department, Private Fundation, Escoles Universitàries Gimbernat, Sant Cugat del Vallès, 08174 Barcelona, Spain; 3Department of Statistics and Operations Research, Universitat Politècnica de Catalunya, 08028 Barcelona, Spain; lourdes.rodero@upc.edu; 4Instituto Poal, 08022 Barcelona, Spain; ingridmoller@gmail.com; 5Basic Sciences Department, Universitat Internacional de Catalunya, 08017 Barcelona, Spain; aperez@uic.es; 6ACTIUM Functional Anatomy Group, 08195 Barcelona, Spain; 7Cattedra di Radiologia “R”-DICMI, Universita di Genova, 16132 Genoa, Italy; mskeletal.radiology@gmail.com

**Keywords:** suprascapular nerve, peripheral nerves, shoulder, dissection, anatomic variation

## Abstract

(1) Background: Suprascapular neuropathy is an important factor contributing to shoulder pain. Given the prevalence of nerve injury and nerve block in the suprascapular notch region, as well as the frequency of arthroscopic procedures on the suprascapular notch, which are recommended in shoulder pain management, its morphology is relevant from a clinical perspective. (2) Methods: Suprascapular nerve course was studied in twelve shoulders by dissection. Its diameter was measured at omohyoid level, proximal to the suprascapular notch and distal to the spinoglenoid notch. A multi-vari chart was used in order to descriptively visualize the results. The variations found were analyzed with a mixed linear model. (3) Results: In two of the six subjects, the suprascapular nerve was divided into two motor branches proximal to the superior transverse scapular ligament. An increase in diameter around the suprascapular notch was detected, with an estimated difference between diameter means of 2.008 mm at the suprascapular notch level and 2.047 mm at the spinoglenoid notch level. (4) Conclusions: A difference in the estimated diameter detected and the fact that the motor branches, which innervate supraspinatus and infraspinatus muscle, were divided proximal to the suprascapular notch may be relevant in the diagnosis and treatment of suprascapular neuropathy and arthroscopic procedures.

## 1. Introduction

The shoulder has complex anatomy and functions with multiple structures, which may be responsible for shoulder pathology [1]. Rotator cuff dysfunction and referred cervical pain are the most common etiologies in primary care [1,2]. However, suprascapular neuropathy is recently becoming an important contributing factor, especially when surgery referral is needed due to pain intensity, muscle atrophy, or poor evolution [3,4,5,6]. Some of these pathologies result in shoulder pain, one of the most prevalent musculoskeletal conditions, representing approximately 16% of all musculoskeletal complaints [7,8].

The suprascapular nerve (SNe) is a mixed sensory and motor nerve arising from the upper trunk of the brachial plexus, and is formed by the ventral rami of C5 and C6 roots and sporadically from the C4 root [9,10,11]. The SNe runs laterally through the posterior cervical triangle behind the omohyoid muscle to the superior border of the scapula travelling across the clavicle. It enters the suprascapular notch (SNo) underneath the superior transverse scapular ligament (STSL), where the suprascapular artery and vein pass over it [9,11,12,13,14]. This SNo region is the most common site of nerve compression and injury along its course [13,15,16], so its morphology is relevant from a clinical point of view. Furthermore, there are multiple anatomic variations (AVs) registered around the SNo region, which can be responsible for SNe entrapment [13,15,16,17]. These AV are also considered a risk factor for neural lesions, though they are not relevant to the kind of neural block or surgical procedures used [18]. Ultrasound-guided regional anesthesia looks for the posterior SN block in the SNo [19,20], and the subomohyoid anterior suprascapular block has recently been studied as another approach to superior trunk block that blocks the majority of shoulder innervation.

Recent evidence about suprascapular neuropathy highlights that the surgical approach through arthroscopy, when is necessary, is recommended because it is minimally invasive and shows good results in terms of pain management and improvement in muscle function and strength [21,22,23,24,25]. Additionally, arthroscopic SNe decompression at the SNo is a technically demanding surgical procedure, as the nerve is located medial to the acromioclavicular joint and requires following several landmarks to reach the area [26]. Both SNe block and arthroscopic SNe decompression at the SNo require extensive knowledge of the SNo and the SNe courses. 

Taking this into account, the main aim of this study is to expand the knowledge of the SNe course, with particular emphasis on the diameter and topography of its branches around the SNo area. 

## 2. Materials and Methods

### 2.1. Anatomical Study and Data Collection

Twelve shoulders from six cryopreserved human cadavers (3 women and 3 men) were bilaterally dissected. The deceased donors were an average of 83 years old (range 68–91 years). All the specimens were from the liberal donations to the Faculty of Medicine and Health Science. They did not present evidence of traumatic injuries or surgical scars. The specimens were coded from A to F according to the dissection order. Subclavian arteries of both specimen F shoulders were injected with black latex to better visualize the arteries.

An anatomical study was performed on each specimen along the course of the SNe and its motor branches in upstanding position with both hands supported in front of the trunk. First, the back skin was removed from the middle line, the trapezius muscle was incised and retracted from medial to lateral to expose the supraspinatus muscle and its anatomical relations. Second, the supraspinatus fossa was observed and adipose tissue, which surrounded the area, was cleaned to expose the SNe proximal and distal to the SNo. Third, the relation with surrounding structures was observed along the SNe course. The nerve was isolated from the vascular-nervous bundle and its diameter was measured at three levels with a Vernier caliper (Mitutoyo ABSOLUTE Solar Caliper Series 500 with ABSOLUTE technology, USA). Fourth, the digital caliper was situated transversally to the SNe as close as possible to the OH belly for the OH level. Fifth, it was placed as close as possible to the STSL approaching from proximal to distal for the SNo level. Finally, it was placed as close as possible to the spinoglenoid notch (SGN) approaching from distal to proximal for the SGN level. Five diameter measurements were carried out at each level by the same person. 

### 2.2. Data Analysis

First, a multi-vari chart was used to descriptively visualize the results in each location of measurement (OHlv, SNolv, and SGNlv). To calculate if there were significant differences between the measured diameter for specimens with and without the AV, a mixed linear model was adjusted for each location to know the effect of the specimen, the right or left side, and having an AV or not. 

A mixed linear model is a parametric statistical model that combines random effects with fixed effects. Mixed-effects modeling allows examination of the condition of interest while simultaneously considering variability within and across subjects. It is especially useful for dealing with different grouping factors such as our specimens, which were considered a random factor and were treated as a block effect, so the variability introduced by the specimen was eliminated to increase the power of the test for AVs. The right or left side and having an AV or not were both considered fixed factors. 

Some Tukey pairwise comparisons were made to study the difference between the diameters at different measurement levels. To compare the data, the mean diameters at each level were estimated for the whole sample and for the specimens with or without the AV.

## 3. Results

### 3.1. Anatomical Study

The SNe emerged under the trapezium from its origin in the brachial plexus. The SNe was observed as it passed dorsally to the posterior belly of the OH muscle running from medial to lateral to the SNo. It went from superior and anterior origins to inferior and posterior positions as it reached the base of the scapular spine. The SNe was also surrounded by adipose tissue and the perineurium, which isolated and protected the nerve from other anatomical structures. The SNe reached the SNo and passed underneath the STSL in 100% of the studied samples. 

In four of the six subjects, the SNe was divided into two motor branches after crossing STSL, but specimens C (Figure 1) and F (Figure 2) presented an AV: the SNe divided into two motor branches proximal to the STSL. In both cases, the medial branch reached the supraspinatus muscle, and the lateral branch continued underneath the inferior transverse scapular ligament to reach the infraspinatus muscle by three to five motor branches.

### 3.2. Diameter Measurements

A mixed linear model was fitted with one random factor (a specimen) and two fixed factors (the left or right side and having an AV or not; the key factors in this study). 

Once the significant effect of the specimen was proved (*p* = 0.087, at the OH level, *p* = 0.096 at the SNo level, *p* = 0.091 at the SGN level), a difference between the left and right side at the OH level (*p* = 0.006) was observed. Additionally, the differences in the SNe diameters between the specimens with and without an AV (*p* = 0.026) were verified.

As there were few specimens, a significance level of 10% was used for the random factor. At the SNo level, there were significant differences in the diameter between the specimens with and without an AV (*p* = 0.001), but there were no differences between the left and right sides (*p* = 0.918). On the other hand, SGN level diameter differences were found between the specimens with and without an AV it (*p* = 0.003) and there were minor differences between the left and right sides (*p* = 0.083). 

Using the results of the fitted model, the differences between diameters at different level could be estimated using the Tukey simultaneous test for differences. The estimated difference between means at the OH level, having an AV or not, was 1.472 mm. At the SNo and SGN levels, a much more significant difference in the mean could be observed, 2.008 and 2.047 mm, respectively, as shown in Table 1. Differences between specimens, the left or right side, and the effect of the AV are presented in Figure 3. 

An evident variability was observed in the measured diameter at the SGN and SNo in specimen C. Significant densification of the tissue, which made the measurement procedure difficult around the nerve, was also observed. 

## 4. Discussion

This is the first study that presents a variation in the diameter of the SNe at the SNo of 2.008 mm ((1.347, 2.670) 95% CI) associated with an AV in which the SNe divided into its motor branches proximal to the STSL.

There is an increasing amount of evidence of AVs of the SNo and its possible relevance to suprascapular neuropathy. Factors such as shape, size, and contents of the SNo and SGN seem to be relevant to the understanding of shoulder pathology, especially when the SNe is involved [13,23,27,28,29,30].

As the SNe contributes to the motor control of the rotator cuff and provides 70% of the sensitive innervation of the glenohumeral and acromioclavicular joints, it may play a relevant role in shoulder pain and dysfunction [14,30,31,32,33,34].

Two anatomic sites of compression, which are separate clinical entities, have been identified: the SNo and the SGN [32]. Both courses involve with deep shoulder posterior pain [31]. Generally, the compression of the SNe at the SNo leads to weakness of both supraspinatus and infraspinatus muscles. In contrast, the compression of the SNe at the SGN leads to isolated infraspinatus weakness [24,32].

A surgical approach is usually used in this neuropathy in case of failure of conservative treatment. The release of the STSL by arthroscopy is considered a less aggressive intervention than an open procedure [21,24,31]. Thus, the relative diameter of the SNe compared with the SNo seems to be a relevant factor both in the possible entrapment of the SNe and in its surgical approach.

Rotator cuff repair and cyst removal procedures by arthroscopy are also commonly used [21,22,23] with better outcomes than open procedures, although they are technically challenging and require an extensive knowledge of SNe and SNo anatomy.

The AV identified in this study may be relevant to SNe entrapment, because it seemed to be accompanied by an increase in the SNe diameter just before passing through the SNo and after leaving the SGN. This variation was also observed by Ebraheim et al. in a cadaveric dissection study in one of the six studied subjects, although diameters were not registered [29].

In addition, one of the subjects who presented the variation (specimen C) also presented an important tissue densification in the SNo area, which complicated the isolation of the SNe for measurement procedures. It has also been documented that scar tissue or densification at the SNo may be relevant to suprascapular neuropathy [22,23].

Although the sample used in this study was small, the average diameter of the SNe measured near the SNo, which was 3.768 (±0.23) mm, is in agreement with that obtained in other recent studies. In the study by Tasaky et al., also carried out by dissection, a mean width of 3.4 (±0.5) mm was recorded in the SNo (*n* = 30) [16]. However, in the study carried out by Polguj et al., whose sample was larger (*n* = 106), a mean diameter of 2.48 (±0.6) mm was recorded [30], which is in line with the average diameter of the 4foursubjects in this study that did not present the AV 2.764 (±0.23) mm.

In the study carried out using ultrasound by Jeziersky et al., the mean diameter of the SNe in the SNo was 3.3 ±1.0 mm, with their sample being the largest studied (*n* = 235) [13]. This result is similar to the mean registered in our sample of 3.713 ± 0.42 mm, which suggests that the AV registered in oud our study could have also be present in their sample. While locating this AV by ultrasound appears to be as challenging, as determining the location of nerve injury in this area is difficult [24], it probably cannot be recorded by ultrasound.

In addition, in this study, a difference in the diameter of the SNe was observed in the SNo, attributable to the right or left side, with the mean diameter being 3.5 ± 1.1 and 1.3 ± 0.4 mm, respectively [13]. In our samples, differences between the sides were observed, especially at the OH level (Figure 3a), none at the SNo level (Figure 3b), and slight differences at the SGN level (Figure 3c). Likewise, the subject who presented the greatest variation between the left and right sides (specimen C in SNo and SGN), presented a marked densification of the tissue around the right side, which was the thickest. The fact that the laterality of the examined subjects is unknown represents an important limitation when assessing whether these differences have any clinical relevance.

The mean diameter of the four extremities that presented the division of the SNe into two branches proximal to the SNo is greater than those recorded so far 4.773 ± 0.23 mm. This thickening prior to the passage of the nerve through one of the areas of major conflict may be a relevant predisposing factor of this variation.

As reported by Jeziersky et al., the mean superior transverse diameter on the SNo is 14.8 ± 4.8 mm and its maximal depth is around 6.3 ± 2.1 mm [13]. A mean difference of 2.008 mm ((1.347, 2.670) 95% CI) proximal to the SNo and 2.047 mm ((1.146, 2.949) 95% CI) distal to the SGN may be a risk factor to consider in suprascapular entrapment neuropathy at the SNo. 

The use of anesthetic block is recommended for both surgical procedures and shoulder pain management. Although the interscalene block is known to be more effective, it carries certain risks at the respiratory level, which can be avoided with the suprascapular block [19,20]. It is recommended to perform the suprascapular block as close as possible to the SSN to cover the largest possible number of sensory branches [28]. Currently, the SNe, the axillary nerve, and the subscapular nerve provide sensory shoulder innervation, with the SNe contributing an estimated 70% of the sensory shoulder innervation [14,35]. This would explain why the interscalene block is preferred to the suprascapular block in surgical procedures [35,36]. However, for the management of shoulder pain, the suprascapular block seems to be equally as effective as the interscalene block, with a lower risk of respiratory complications or neurological symptoms in the arm [19,20,35]. By considering the increasingly widespread use of the suprascapular block for the management of shoulder pain, expanding the knowledge of the possible AV that occurs at the SNo level is of clinical interest.

Given the limited size of our sample, the real incidence of this AV is unknown. In studies carried out on cadavers [16,29,30], only Ebraheim et al. documented the same AV. The largest sample [13] was studied by ultrasound, which can only detect the thickening of the nerve, not the AV. Considering that the mean diameter of the sample studied by Jeziersky et al. is similar to that estimated in our sample, this variation may be relatively frequent.

The mixed linear model used to analyze our sample was adjusted to eliminate the variability introduced by the specimens to increase the power of the test for the AV. As shown, the effect of the AV was statistically significant, so having a larger sample size is required to show its effect more clearly. 

Authors should discuss the results and how they can be interpreted from the perspectives of previous studies and of the working hypotheses. The findings and their implications should be discussed in the broadest context possible. Future research directions may also be highlighted.

## 5. Conclusions

Shoulder pain has a variety of potential causes, including suprascapular neuropathy, which is still considered a diagnosis of exclusion. The AV registered in this study may be clinically relevant in SNe entrapment neuropathies as it involves an increase in diameter of the SNe at the SNo. The finding that the motor branches, which innervate the supraspinatus and infraspinatus muscles, were divided proximal to the SNo, may be a distortion factor in the suprascapular block or arthroscopic procedures that should be considered. 

## Figures and Tables

**Figure 1 ijerph-19-07065-f001:**
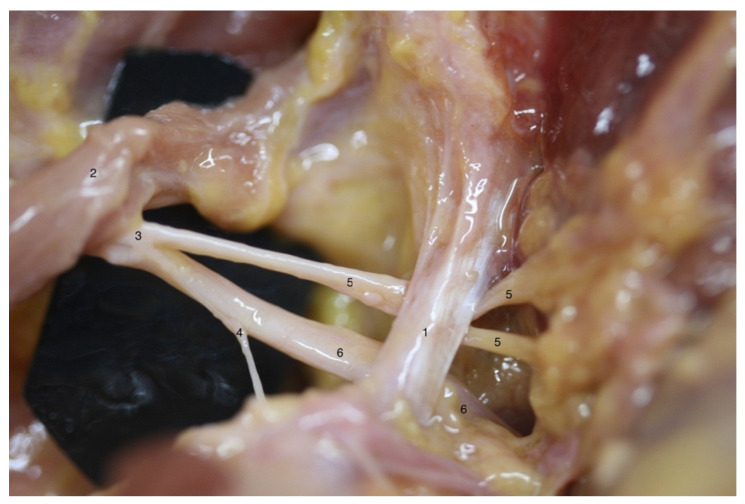
Anatomical variation on left shoulder from specimen C: different anatomical structures could be observed as follows: (1) superior transverse scapular ligament; (2) inferior belly of omohyoid muscle; (3) suprascapular nerve; (4) acromioclavicular sensitive branch; (5) supraspinatus motor branch; (6) infraspinatus motor branch; note that the division into supraspinatus and infraspinatus motor branches is proximal to superior transverse scapular ligament.

**Figure 2 ijerph-19-07065-f002:**
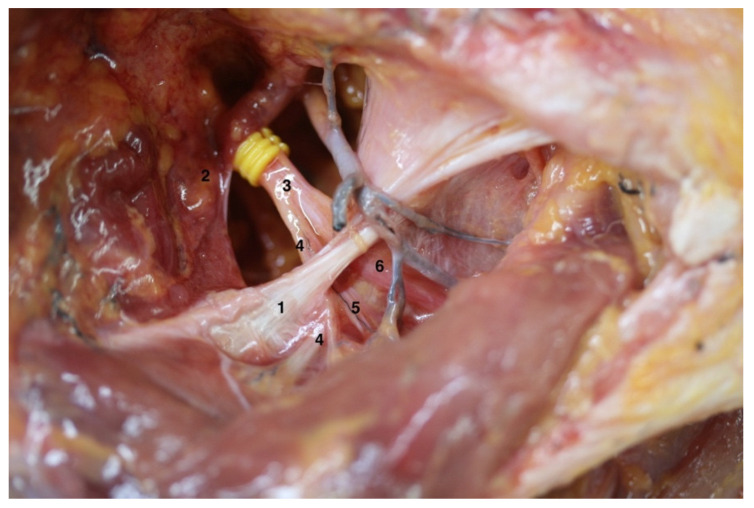
Anatomical variation on the right shoulder from specimen F: the different anatomical structures could be observed as follows: (1) Superior transverse scapular ligament; (2) Inferior belly of the omohyoid muscle; (3) Suprascapular nerve; (4) Acromioclavicular sensitive branch; (5) Supraspinatus motor branch; (6) Infraspinatus motor branch; note that the division into the supraspinatus and infraspinatus motor branches is proximal to superior transverse scapular ligament.

**Figure 3 ijerph-19-07065-f003:**
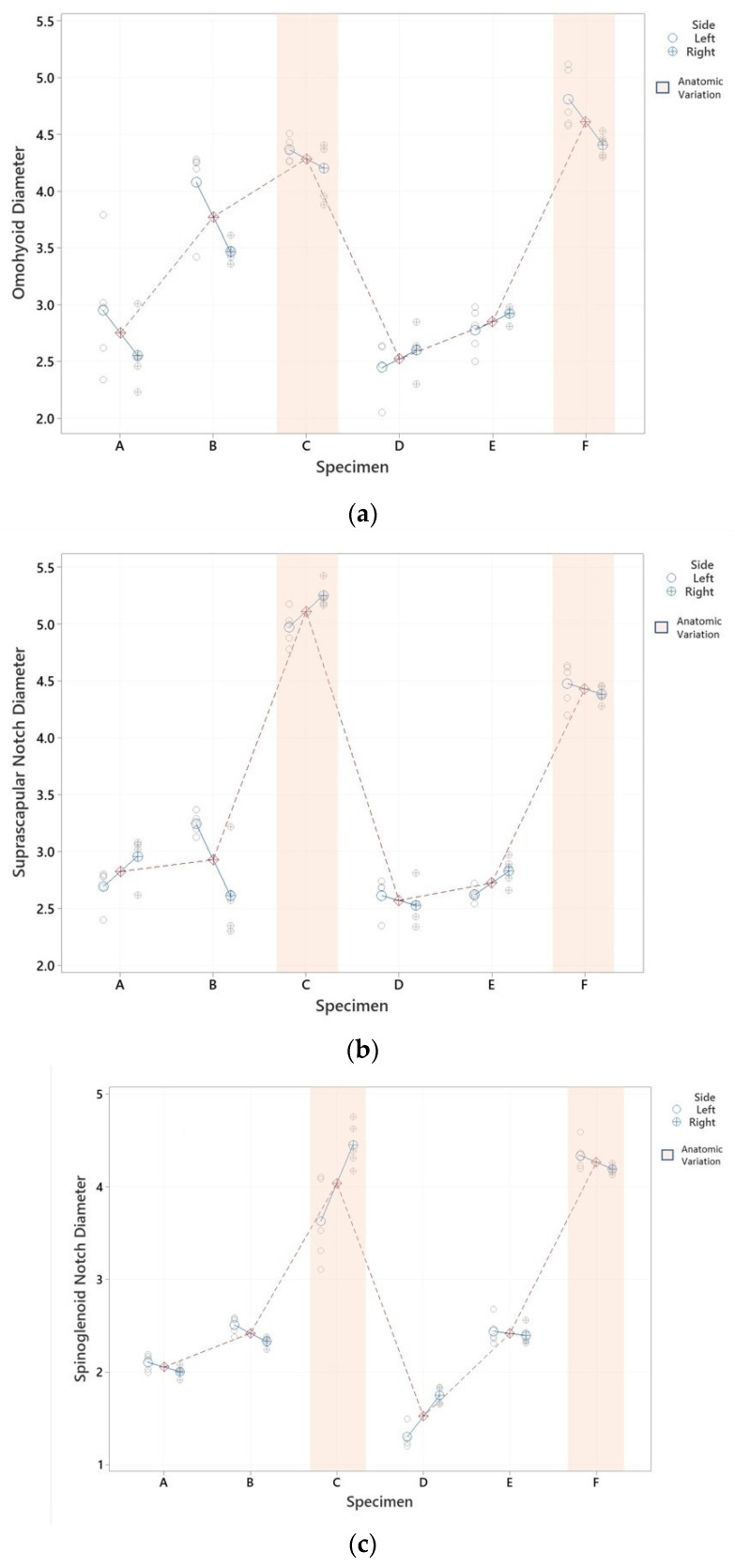
SNe diameter graphical representation by specimens. (**a**) SNe diameter graphical representation by specimens at OHlv; (**b**) SNe diameter graphical representation by specimens at SNolv; (**c**) SNe diameter graphical representation by specimens at SGNlv. Differences between specimens (each red point is the mean of its diameter) are represented in OHlv (**a**), SNolv (**b**), and SGNlv diameters (**c**). Differences between sides (each blue point is the mean of left and right sides), as well as the effect of the AV (marked in orange), were observed in 5 measurements.

**Table 1 ijerph-19-07065-t001:** SN diameter measurement. The estimated mean diameter on the different levels were observed (OHlv, omohyoid level; SNolv, suprascapular notch level; SGNlv, spinoglenoid notch level) for the whole sample (SM, sample mean), AV group (VM, variation mean), and no-AV group (noVM, no variant mean) with standard deviation. The estimated difference of means (DM) between VM and noVM with its corresponding confidence level and adjusted *p*-value are noted.

Level	SM	VM	noVM	DM	Simultaneous 95% CI	Adjusted *p*-Value
**OHlv**	3.713 ± 0.42	4.449 ± 0.42	2.977 ± 0.42	1.472	(0.293, 2.650)	0.026
**SNolv**	3.768 ± 0.23	4.773 ± 0.23	2.764 ± 0.23	2.008	(1.347, 2.670)	0.001
**SGNlv**	3.130 ± 0.32	4.153 ± 0.32	2.106 ± 0.32	2.047	(1.146, 2.949)	0.003

Individual confidence level = 95.00%.

## Data Availability

Not applicable.

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
