# Peer review of "Variations in the Course and Diameter of the Suprascapular Nerve: Anatomical Study"

_ijerph, 2022, doi:10.3390/ijerph19127065_

Round 1

Reviewer 1 Report

The main flaws of this study is small number of specimens (according to the cited literature) and focusing only on issues connected with arthroscopic release of SSN.

Although it's relevant value is good description of anatomical variances of the nerve it doesn't mention vascular variations - present in another studies.

Some issues to clarify/correct:

- using AV or VM - for anatomical variations - taht should be unified along the text

- SN and SSN is used through the text misleading the reader if scapular notch or suprascapular nerve is concerned

- 184 and 245 lines provide contractionary information about sensory supply of glenohumeral joint

- in line 255 the sentence "The largest sample [13], was studied with" is missing the ending 

Reviewer 2 Report

1: Twelve samples are enough for this type of study?

2: : Data Analysis subsection should be added in
      the materials and method section

3: Statistics analysis method should be explicitly

4: Was a parametric or no-parametric statistics method used in this study

5: Is there a reason for the used subject age in this study?

Author Response

Thank you

Round 2

Reviewer 1 Report

Authors clarified my remarks.